# The Emergence of Individuality in Multi-Agent Reinforcement Learning

## Abstract

Individuality is essential in human society. It induces the division of labor and thus improves the efficiency and productivity. Similarly, it should also be a key to multi-agent cooperation. Inspired by that individuality is of being an individual separate from others, we propose a simple yet efficient method for the emergence of individuality (EOI) in multi-agent reinforcement learning (MARL). EOI learns a probabilistic classifier that predicts a probability distribution over agents given their observation and gives each agent an intrinsic reward of being correctly predicted by the classifier. The intrinsic reward encourages the agents to visit their own familiar observations, and learning the classifier by such observations makes the intrinsic reward signals stronger and in turn makes the agents more identifiable. To further enhance the intrinsic reward and promote the emergence of individuality, two regularizers are proposed to increase the discriminability of the classifier. We implement EOI on top of popular MARL algorithms. Empirically, we show that EOI outperforms existing methods in a variety of multi-agent cooperative scenarios.

## 1 Introduction

Humans develop into distinct individuals due to both genes and environments (Freund et al., 2013). Individuality induces the division of labor (Gordon, 1996), which improves the productivity and efficiency of human society. Analogically, the emergence of individuality should also be essential for multi-agent cooperation.

Although multi-agent reinforcement learning (MARL) has been applied to multi-agent cooperation, it is widely observed that agents usually learn similar behaviors, especially when the agents are homogeneous with shared global reward and co-trained (McKee et al., 2020). For example, in multi-camera multi-object tracking (Liu et al., 2017), where camera agents learn to cooperatively track multiple objects, the camera agents all tend to track the easy object. However, such similar behaviors can easily make the learned policies fall into local optimum. If the agents can respectively track different objects, they are more likely to solve the task optimally. Many studies formulate such a problem as task allocation or role assignment (Sander et al., 2002; Dastani et al., 2003; Sims et al., 2008). However, they require that the agent roles are rule-based and the tasks are pre-defined, and thus are not general methods. Some studies intentionally pursue difference in agent policies by diversity (Lee et al., 2020; Yang et al., 2020) or by emergent roles (Wang et al., 2020a), however, the induced difference is not appropriately linked to the success of task. On the contrary, the emergence of individuality along with learning cooperation can automatically drive agents to behave differently and take a variety of roles, if needed, to successfully complete tasks.

Biologically, the emergence of individuality is attributed to innate characteristics and experiences. However, as in practice RL agents are mostly homogeneous, we mainly focus on enabling agents to develop individuality through interactions with the environment during policy learning. Intuitively, in multi-agent environments where agents respectively explore and interact with the environment, individuality should emerge from what they experience. In this paper, we propose a novel method for the emergence of individuality (EOI) in MARL. EOI learns a probabilistic classifier that predicts a probability distribution over agents given their observation and gives each agent an intrinsic reward of being correctly predicted probability by the classifier. Encouraged by the intrinsic reward, agents tend to visit their own familiar observations. Learning the probabilistic classifier by such observations makes the intrinsic reward signals stronger and in turn makes the agents more identifiable. In this

closed loop with positive feedback, agent individuality emerges gradually. However, at early learning stage, the observations visited by different agents cannot be easily distinguished by the classifier, meaning the intrinsic reward signals are not strong enough to induce agent characteristics. Therefore, we propose two regularizers for learning the classifier to increase the discriminability, enhance the feedback, and thus promote the emergence of individuality.

EOI is compatible with centralized training and decentralized execution (CTDE). We realize EOI on top of two popular MARL methods, MAAC (Iqbal & Sha, 2019) and QMIX (Rashid et al., 2018). For MAAC, as each agent has its own critic, it is convenient to shape the reward for each agent. For QMIX, we introduce an auxiliary gradient and update the individual value function by both minimizing the TD error of the joint action-value function and maximizing its cumulative intrinsic rewards. We evaluate EOI in three scenarios where agents are preferred to take different roles, *i.e.*, Pac-Men, Windy Maze, and Firefighters, and we empirically demonstrate that EOI significantly outperforms existing methods. Additionally, in a micro-task of StarCraft II (Samvelyan et al., 2019) where the need for the division of labor is unknown, EOI also learns faster than existing methods. By ablation studies, we confirm that the proposed regularizers indeed improve the emergence of individuality even if agents have the same innate characteristics.

## 2 RELATED WORK

**MARL.** We consider the formulation of Decentralized Partially Observable Markov Decision Process (Dec-POMDP), where at each timestep $t$ each agent $i$ receives a local observation $o_i^t$, takes an action $a_i^t$, and gets a shared global reward $r^t$. Agents together aim to maximize the expected return $\mathbb{E} \sum_{t=0}^{T} \gamma^t r^t$, where $\gamma$ is a discount factor and $T$ is the time horizon. Many methods have been proposed for Dec-POMDP, most of which adopt CTDE. Some methods (Lowe et al., 2017; Foerster et al., 2018; Iqbal & Sha, 2019) extend policy gradient into multi-agent cases. Value function factorization methods (Sunehag et al., 2018; Rashid et al., 2018; Son et al., 2019) decompose the joint value function into individual value functions. Communication methods (Das et al., 2019; Jiang et al., 2020) share information between agents for better cooperation.

**Behavior Diversification.** Many cooperative multi-agent applications require agents to take different behaviors to complete the task successfully. Behavior diversification can be handcrafted or emerge through agents' learning. Handcrafted diversification is widely studied as task allocation or role assignment. Heuristics (Sander et al., 2002; Dastani et al., 2003; Sims et al., 2008; Macarthur et al., 2011) assign specific tasks or pre-defined roles to each agent based on goal, capability, visibility, or by search. M$^3$RL (Shu & Tian, 2019) learns a manager to assign suitable sub-tasks to rule-based workers with different preferences and skills. These methods require that the sub-tasks and roles are pre-defined, and the worker agents are rule-based. However, in general, the task cannot be easily decomposed even with domain knowledge and workers are learning agents.

The emergent diversification for single agent has been studied in DIAYN (Eysenbach et al., 2019), which learns reusable diverse skills in complex and transferable tasks without any reward signal by maximizing the mutual information between states and skill embeddings as well as entropy. In multi-agent learning, SVO (McKee et al., 2020) introduces diversity into heterogeneous agents for more generalized and high-performing policies in social dilemmas. Some methods are proposed for behavior diversification in multi-agent cooperation. ROMA (Wang et al., 2020a) learns a role encoder to generate the role embedding, and learns a role decoder to generate the neural network parameters. However, there is no mechanism that guarantees the role decoder can generate different parameters, taking as input different role embeddings. Learning low-level skills for each agent using DIAYN is considered in Lee et al. (2020); Yang et al. (2020), where agents' diverse low-level skills are coordinated by the high-level policy. However, the independently trained skills limit the cooperation, and diversity is not considered in the high-level policy.

## 3 METHOD

Individuality is of being an individual separate from others. Motivated by this, we propose EOI, where agents are intrinsically rewarded in terms of being correctly predicted by a probabilistic classifier that is learned based on agents' observations. If the classifier learns to accurately distinguish agents, agents should behave differently and thus individuality emerges. Two regularizers are introduced for

learning the classifier to enhance intrinsic reward signal and promote individuality. Unlike existing work on behavior diversification, EOI directly correlates individuality with the task by intrinsic reward, and thus individuality emerges naturally during agents' learning. EOI can be applied to Dec-POMDP tasks and trained along with CTDE algorithms. We design practical techniques to implement EOI on top of two popular MARL methods, MAAC and QMIX. We also propose an unbiased gradient estimation for learning from other agents' experiences to accelerate learning, which appears in Appendix A.1.

### 3.1 INTRINSIC REWARD

As illustrated in Figure 1, two agents learn to collect two items, where item $b$ is the easier one. Collecting each item, they will get a global reward of $+1$. Sequentially collecting the two items by one agent or together is sub-optimal, sometimes impossible when the task has a limited horizon. The optimal solution is that the two agents go collecting different items simultaneously. It is easy for both agents to learn to collect the easier item $b$. However, after that, item $a$ becomes even harder to be explored, and the learned policies easily fall at local optimum. Nevertheless, the emergence of individuality can address the problem, *e.g.*, agents prefer to collect different items.

To enable agents to develop individuality, EOI learns a probabilistic classifier $P(I|O)$ to predict a probability distribution over agents given on their observation, and each agent takes the correctly predicted probability as the intrinsic reward at each timestep. Thus, the reward function for agent $i$ is modified as

$$r + \alpha p(i|o_i), \tag{1}$$

where $r$ is the global environmental reward, $p(i|o_i)$ is the predicted probability of agent $i$ given its observation $o_i$, and $\alpha$ is a tuning parameter to weight the intrinsic reward. With the reward shaping, EOI works as follows. If there is initial difference between agent policies in terms of visited observations, the difference is captured by $P(I|O)$ as it is fitted using agents'

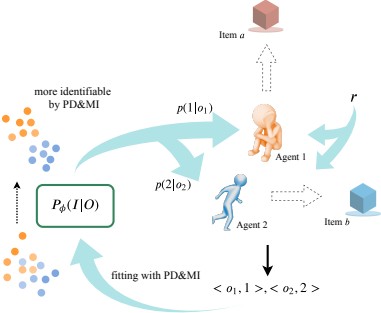

Figure 1: EOI

experiences. The difference is then fed back to each agent as an intrinsic reward. As agents maximize the expected return, the difference in agents' policies is exacerbated together with optimizing the environmental return. Therefore, the learning process is a closed loop with positive feedback. As agents progressively behave more identifiably, the classifier can distinguish agents more accurately, and thus individuality emerges gradually.

The classifier $P_\phi(I|O)$ is parameterized by a neural network $\phi$ and learned in a supervised way. At each timestep, we take each agent $i$'s observation $o_i$ as input and the agent index $i$ as the label and store the pair $< o_i, i >$ into a buffer $\mathcal{B}$. $\phi$ is updated by minimizing the cross-entropy loss (CE), which is computed based on the uniformly sampled batches from $\mathcal{B}$.

### 3.2 REGULARIZERS OF $P_\phi(I|O)$

In the previous section, we assume that there is some difference between agents' policies. However, in general, the difference between initial policies is small (even no differences if agents' policies are initially by the same network weights), and the policies will quickly learn similar behaviors as in the example in Figure 1. Therefore, the intrinsic rewards are nearly the same for each agent, which means no feedback in the closed loop. To generate the feedback in the closed loop, the observation needs to be identifiable and thus the agent can be distinguished in terms of observations by $P_\phi(I|O)$. To address this, we propose two regularizers: positive distance (PD) and mutual information (MI) for learning $P_\phi(I|O)$.

**Positive Distance.** The positive distance is inspired from the triplet loss (Schroff et al., 2015) in face recognition, which is proposed to learn identifiable embeddings. Since $o_i^t$ and its previous observations $\{o_i^{t-\Delta t}, o_i^{t-\Delta t+1}, \cdots, o_i^{t-1}\}$ are distributed on the trajectory generated by agent $i$, the previous observations in the $\Delta t$-length window could be seen as the positives of $o_i^t$. To make the probability distribution on the anchor $o_i^t$ close to that on the positives, we sample an observation $o_i^{t'}$

from $\{o_i^{t-\Delta t}, o_i^{t-\Delta t+1}, \cdots, o_i^{t-1}\}$ and minimize the cross-entropy loss

$$\text{CE}\left(p_\phi(\cdot|o_i^t), p(\cdot|o_i^{t'})\right). \tag{2}$$

The positive distance minimizes the intra-distance between the observations with the same "identity", which hence enlarges the margin between different "identities". As a result, the observations become more identifiable. Since the positives are naturally defined on the trajectory, the identifiability generated by the positive distance is actually induced by the agent policy. Defining the negatives is hard but we find that just using the positive distance works well in practice.

**Mutual Information.** If the observations are more identifiable, it is easier to infer the agent that visits the given observation most, which indicates the higher mutual information between the agent index and observation. Therefore, to further increase the discriminability of the classifier, we maximize their mutual information,

$$\text{MI}(I;O) = \mathcal{H}(I) - \mathcal{H}(I|O)$$
$$= \mathcal{H}(I) - \mathbb{E}_{o\sim p(o)}\left[\sum_i -p(i|o)\log p(i|o)\right]. \tag{3}$$

Since we store $< o_i, i >$ of every agent in $\mathcal{B}$, the number of samples for each agent is equal. Fitting $P_\phi(I|O)$ using batches from $\mathcal{B}$ ensures $\mathcal{H}(I)$ is a constant. To maximize $\text{MI}(I;O)$ is to minimize $\mathcal{H}(I|O)$. Therefore, equivalently, we sample batches from $\mathcal{B}$ and minimize

$$\text{CE}\left(p_\phi(\cdot|o_i^t), p_\phi(\cdot|o_i^t)\right). \tag{4}$$

Therefore, the optimization objective of $P_\phi(I|O)$ is to minimize

$$\text{CE}\left(p_\phi(\cdot|o_i^t), \text{one\_hot}(i)\right) + \beta_1 \text{CE}\left(p_\phi(\cdot|o_i^t), p(\cdot|o_i^{t'})\right) + \beta_2 \text{CE}\left(p_\phi(\cdot|o_i^t), p_\phi(\cdot|o_i^t)\right), \tag{5}$$

where $\beta_1$ and $\beta_2$ are hyperparameters. The regularizers increase the discriminability of $P_\phi(I|O)$, make the intrinsic reward signals stronger to stimulate the agents to be more distinguishable, and eventually promote the emergence of individuality. In this sense, $P_\phi(I|O)$ not only is the posterior statistics, but also serves as the inductive bias of agents' learning. The learning process of EOI is illustrated in Figure 1.

### 3.3 IMPLEMENTATION WITH MAAC AND QMIX

Existing methods for reward shaping in MARL focus on independent learning agents *e.g.*, McKee et al. (2020); Wang et al. (2020b); Du et al. (2019). How to shape the reward in Dec-POMDP for centralized training has not been deeply studied. Since the intrinsic reward can be exactly assigned to the specific agent, individually maximizing the intrinsic reward is more efficient than jointly maximizing the sum of all agents' intrinsic rewards (Hughes et al., 2018). Adopting this idea, we respectively present the implementation with MAAC and QMIX for realizing EOI. MAAC is an off-policy actor-critic algorithm, where each agent learns its own critic, thus it is convenient to directly give the shaped reward $r + \alpha p_\phi(i|o_i)$ to the critic of each agent $i$, without modifying other components, as illustrated in Figure 2(a). The TD error of the critic and the policy gradient are the same as in MAAC (Iqbal & Sha, 2019).

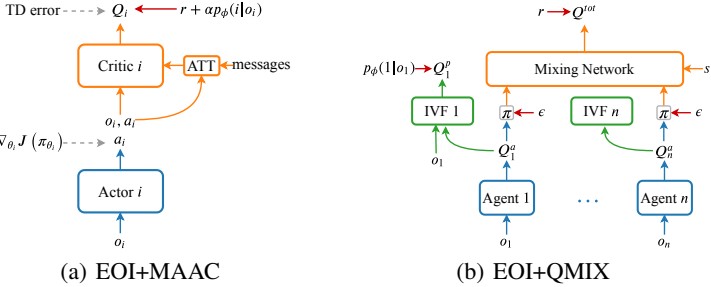

(a) EOI+MAAC                  (b) EOI+QMIX

Figure 2: Illustration of EOI with MAAC and QMIX.

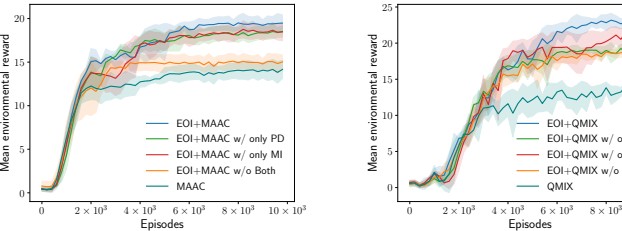

Figure 3: Illustration of scenarios: Pac-men (*left*), Windy Maze (*mid*), Firefighters (*right*).

In QMIX, each agent $i$ has an individual action-value function $Q_i^a$. All the individual action-value functions are monotonically mixed into a joint action-value $Q^{tot}$ by a mixing network. Each agent selects the action with the highest individual value, but the individual value has neither actual meaning nor constraints (Rashid et al., 2018). Therefore, we can safely introduce an auxiliary gradient of the intrinsic reward to the individual action-value function $Q_i^a$. Each agent $i$ learns an intrinsic value function (IVF) $Q_i^p$, which takes as input the observation $o_i$ and the individual action-value vector $Q_i^a(o_i)$ and approximates $\mathbb{E} \sum_{t=0}^{T} \gamma^t p(i|o_i^t)$ by minimizing the TD error,

$$\mathbb{E}_{<o_i, Q_i^a(o_i), o_i'>\sim\mathcal{D}} \left[ (Q_i^p (o_i, Q_i^a(o_i)) - y)^2 \right], \text{ where } y = p_\phi(i|o_i) + \gamma \bar{Q}_i^p(o_i', \bar{Q}_i^a(o_i')). \quad (6)$$

$\bar{Q}_i^a$ and $\bar{Q}_i^p$ are the target value functions and $\mathcal{D}$ is the replay buffer. In order to improve both the global reward and intrinsic reward, we update $Q_i^a$, parameterized by $\theta_i$, towards maximizing $\mathbb{E}\left[Q_i^p(o_i, Q_i^a(o_i; \theta_i))\right]$ along with minimizing the TD error of $Q^{tot}$ (denoted as $\delta^{tot}$), as illustrated in Figure 2(b). Since the intrinsic value function is differentiable with respect to the individual action-value vector $Q_i^a(o_i; \theta_i)$, we can establish the connection between $Q_i^p$ and $Q_i^a$ by the chain rule, and the gradient of $\theta_i$ is,

$$\nabla_{\theta_i} J(\theta_i) = \frac{\partial \delta^{tot}}{\partial \theta_i} - \alpha \frac{\partial Q_i^p(o_i, Q_i^a(o_i; \theta_i))}{\partial Q_i^a(o_i; \theta_i)} \frac{\partial Q_i^a(o_i; \theta_i)}{\partial \theta_i}. \quad (7)$$

MAAC and QMIX are off-policy algorithms, and the environmental rewards are stored in the replay buffer. However, the intrinsic rewards are recomputed in the sampled batches before each update, since $P_\phi(I|O)$ is co-evolving with the learning agents, the previous intrinsic reward is outdated. The joint learning process of the classifier and agent policies can be mathematically formulated as a bi-level optimization, which is detailed in Appendix A.2.

## 4 EXPERIMENTS

To clearly interpret the mechanism of EOI, we design three scenarios: Pac-Men, Windy Maze, and Firefighters, which are illustrated in Figure 3, and adopt a micro-task of StarCraft II (Samvelyan et al., 2019). In the evaluation, we first verify the effectiveness of EOI and the two regularizers on both MAAC and QMIX by ablation studies. Then, we compare EOI against EDTI (Wang et al., 2020b) to verify the advantages of individuality with sparse reward, and against ROMA (Wang et al., 2020a) and Lee et al. (2020); Yang et al. (2020) (denoted as HC and HC-simul respectively) to investigate the advantages of individuality over emergent roles and diversity. The details about the experimental settings and the hyperparameters are available in Appendix A.3.

### 4.1 PAC-MEN

There are four pac-men (agents) initialized at the maze center, and some randomly initialized dots, illustrated in Figure 3 (*left*). Each agent has a local observation that contains a square view with $5 \times 5$

Figure 4: Learning curves in Pac-Men: EOI+MAAC (*left*) and EOI+QMIX (*right*).

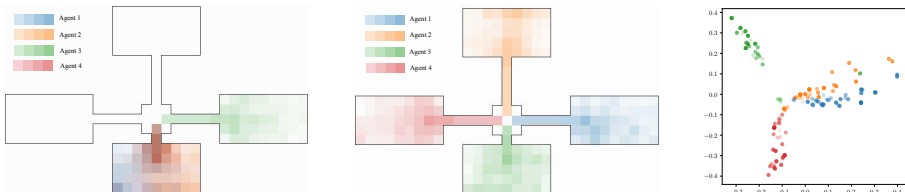

Figure 5: Distributions of agents' positions of QMIX (*left*) and EOI+QMIX (*mid*), and kernel PCA of agents' observations of EOI+QMIX (*right*) in Pac-Men. The darker color means the higher value.

grids centered at the agent itself. At each timestep, each agent can move to one of four neighboring grids or eat a dot. The agents only get a global reward, *i.e.*, the total eaten dots, at the final timestep.

As depicted Figure 4, MAAC and QMIX get the lowest mean environmental reward respectively, since some agents learn to go to the same room and compete for the dots. This can be verified by the position distribution of QMIX agents in Figure 5 (*left*), where three agents move to the same room. MAAC agents behave similarly. At the early stage of training, it is easy for the agents to explore the bottom room and eat dots there to improve the reward. Once the agents learn such policies, it is hard to explore other rooms, so the agents learn similar behaviors and fall at the local optimum.

Driven by the intrinsic reward without both regularizers, EOI obtains better performance than MAAC and QMIX. But the improvement is not significant since the observations of different agents cannot be easily distinguished when there is little difference between initial policies. The regularizers of PD and MI can increase the discriminability of $P_\phi(I|O)$, providing stronger intrinsic signals. Guided by $P_\phi(I|O)$ with PD or MI, the agents go to different rooms and eat more dots. MI theoretically increases the discriminability even the initial policies have no differences, while PD makes the observations distinguishable according to policies. Combining the advantages of the two regularizers leads to higher and steadier performance, as shown in Figure 4. With both two regularizers, the agents respectively go to the four rooms and achieve the highest reward, which is indicated in Figure 5 (*mid*). We also visualize the observations of different agents by kernel PCA, as illustrated in Figure 5 (*right*), where darker color means higher correctly predicted probability of $P_\phi(I|O)$. We can see $P_\phi(I|O)$ can easily distinguish agents given their observations.

To further investigate the effect of the regularizers, we show the learning curves of intrinsic reward and environmental reward of EOI with or without regularizers in Figure 6. EOI with the regularizers converges to higher intrinsic reward than that without regularizers, meaning agents behave more distinctly. With regularizers, the rapid increase of intrinsic reward occurs before that of environmental reward, which indicates the regularizers make $P_\phi(I|O)$ also serve as the inductive bias for the emergence of individuality.

We also investigate the influence of the difference between initial policies. The action distributions (over visited states) of the four initial action-value functions in QMIX are illustrated in Figure 7, where we find there is a large difference between them. The inherent observation makes agents distinguishable initially, which is the main reason EOI without the regularizers works well. We then initiate the four action-value functions with the same network weights and re-run the experiments. The performance of EOI without the regularizers drops considerably, while EOI with the regularizers has almost no difference, as illustrated in Figure 8. Therefore, we can conclude that PD and MI also make the learning more robust to the initial policies. In other words, even with the same innate characteristics, agent individuality still emerges.

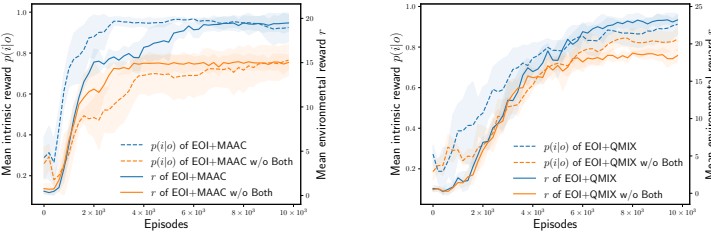

Figure 6: Learning curves of intrinsic reward and environmental reward of EOI+MAAC (*left*) and EOI+QMIX (*right*) in Pac-Men.

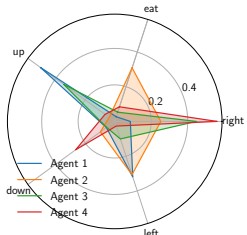

Figure 7: Action distributions of the initial action-value functions in QMIX.

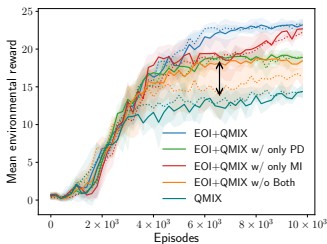

Figure 8: Learning curves with EOI+QMIX. The dotted lines are the version with the same initial action-value functions.

## 4.2 WINDY MAZE

In the scenario, there are two pirates (agents) initialized at the bottom of the T-shaped maze, and two treasure boxes initialized at the two ends, as illustrated in Figure 3 (*mid*). Each agent has a local observation that contains a square view with $5 \times 5$ grids and could move to one of four neighboring grids or open the box. There is a wind running right from the dotted line. Shifted by the wind, forty percent of the time, the agent will move to the right grid whichever action it takes. The agents could only get a global reward, *i.e.*, the number of the opened boxes, at the final timestep.

Figure 9 shows the learning curves of EOI+MAAC and EOI+QMIX. Under the effect of the wind, it is easy to go to the right end and open the box, even if the agent acts randomly. Limited by the time horizon, it is impossible that the agent first opens the right box and then goes to the left end for the other box. MAAC and QMIX only achieve the mean reward of $1$. Due to the wind and the small state space, the trajectories of the two agents are similar, thus EOI without regularizers provides little help, where the intrinsic reward is not strong enough to induce individuality. With regularizers, the observations on the right path and the left path can be discriminated gradually. In the learning process, we find that the agents will first both learn to open the right box. Then one of them will change its policy and go left for higher intrinsic reward, and eventually the agents develop the distinct policies and get a higher mean reward.

The left box is hard to explore, thus we also compare EOI with EDTI (Wang et al., 2020b), a multi-agent exploration method. EDTI gives the agent an intrinsic motivation, considering both curiosity and influence. It rewards the agent to explore the rarely visited states and to maximize the influence on the expected returns of other agents. The optimization of the intrinsic motivation is integrated with PPO (Schulman et al., 2017). However, EDTI does not perform well in this scenario. When the agent policies are similar, the two agents will be curious about similar states. When they both explore novel states, the environmental reward decreases, which draws the agents to go back to the right box again. So the curiosity is not stable, and we can see its fluctuation during training in Figure 9.

## 4.3 FIREFIGHTERS

There are some burning grids in two areas and two rivers in other areas. Four firefighters (agents) are initiated at the same burning area, illustrated in Figure 3 (*right*). The agent has a local observation that contains a square view with $5 \times 5$ grids and can move to one of four neighboring grids, spray water, or pump water. They share a water tank, which is initiated with four units water. Once the agent sprays water on the burning grid, it puts out the fire and consumes one unit water. Once the agent pumps water in the river, the water in the tank increases by one unit. The agents only get a global reward, *i.e.*, the number of the extinguished burning grids, at the final timestep.

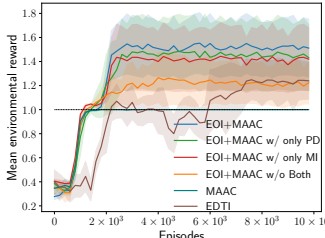
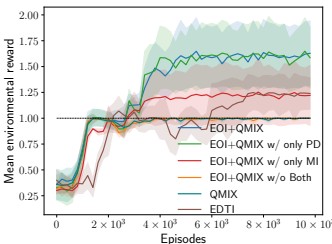

Figure 9: Learning curves in Windy Maze: EOI+MAAC (*left*) and EOI+QMIX (*right*).

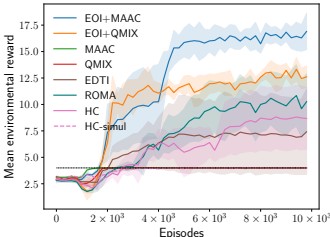 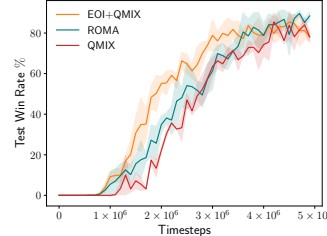

Figure 10: Learning curves in Firefighters (*left*) and 8m (*right*).

As illustrated by Figure 10 (*left*), MAAC and QMIX fall into local optimum. The agents learn to put out the fire around the initial position until the water is exhausted, and get the mean reward 4. They do not take different roles of pumping water or putting out the fire farther. Benefited from the emergent individuality, EOI+MAAC achieves the best performance with a clear division of labor, where two agents go to the river for pumping water, and two agents go to different areas for fighting with fire. EDTI could escape from the local optimum, but it converges to a lower mean reward than EOI. This is because encouraging curiosity and influence cannot help division of labor, and the intrinsic motivation of EDTI is too complex to suit the cases with more than two agents.

ROMA (Wang et al., 2020a), based on QMIX, uses a role encoder taking the local observation to generate role embedding, and decodes the role to generate the parameters of individual action-value function in QMIX. However, ROMA converges to a lower mean reward than EOI+QMIX and learns much slower. Due to the large parameter space, generating various parameters for the emergent roles is less efficient than encouraging diverse behaviors by reward shaping. Moveover, since the agents are initiated with the same observations, the role encoder will generate the similar role embeddings, which means similar agent behaviors at the beginning, bringing difficulty to the emergent roles.

HC (Lee et al., 2020) first provides each agent a set of diverse skills, which are learned independently by DIAYN (before $6 \times 10^3$ episodes), then learns a high-level policy to select the skill for each agent. Since the agents are trained independently without coordinating with others' skills, the skills could be diverse but might not be useful, *e.g.*, ignoring the pumping action. So it converges to a lower mean reward. We also train the primitive skills and high-level policy simultaneously, *i.e.*, HC-simul (Yang et al., 2020), and find it falls into local optimum. Before the diversity emerges in the skills, the high-level policy learns to choose the skill of putting out the nearby fire for the cheap reward, however other skills would not be chosen to learn diversity. That is to say, HC-simul only considers diversity in the low-level policies but ignores that in the high-level policy. EOI encourages individuality with the coordination provided by centralized training, considering both of the global reward and other agents' policies. Therefore, EOI is more effective than HC and HC-simul. Moreover, EOI also outperforms ROMA and HC in Pac-Men, available in Appendix A.4.

### 4.4 STARCRAFT II

We additionally evaluate EOI on a micromanagement scenario of StarCraft II (Samvelyan et al., 2019) 8m, where 8 Marines face 8 enemy Marines and the need for the division of labor is unknown. As illustrated in Figure 10 (*right*), EOI learns faster than ROMA and QMIX, showing the classifier $P_\phi(I|O)$ could be effectively trained and provide useful intrinsic signals in the highly dynamic environment with many agents. Note that we represent the individual action-value function as MLP, not GRU, which means that the action value is generated only based on $o_i^t$, and the agents do not share the neural network parameters so that the result of QMIX is different from that in the original paper (Rashid et al., 2018).

## 5 CONCLUSION

We have proposed EOI, a novel method for the emergence of individuality in MARL. EOI learns a probabilistic classifier that predicts a probability distribution over agents given their observation and gives each agent an intrinsic reward of being correctly predicted by the classifier. Two regularizers are introduced to increase the discriminability of the classifier. We realized EOI on top of two popular MARL methods and empirically demonstrated that EOI outperforms existing methods in a variety of multi-agent cooperative tasks.

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

## A   APPENDIX

### A.1   LEARNING FROM EXPERIENCES OF OTHER AGENTS

When agents are exploring the environment, agents have various exploration patterns. The experiences of one agent can be helpful for the learning of other agents, *i.e.*, improving data efficiency and accelerating learning. Therefore, we propose an unbiased gradient estimate for learning from other agents' experiences. In centralized training, all agents' experience tuples $< O^t, A^t, r^t, O^{t+1}, s^t, s^{t+1} >$ are stored in the replay buffer, and sampled as batch $\Omega$ to compute the gradients jointly. For each agent $i$, the probability of sampled observation is $p(o|i)$. To use others' experiences, we randomly rearrange the agent index in a subset of $\Omega$, denoted as $\omega$. For example, in a two-agent case, two agents exchange $o^t$, $a^t$, and $o^{t+1}$ to build a new training experience. In the experiences with shuffled index $\omega$, the training experience of each agent $i$ is sampled based on $p(o)$ rather than $p(o|i)$. By Bayes rule,

$$p(o|i) = \frac{p(i|o)p(o)}{p(i)}.$$

Since the number of observations for each agent is equal in the training set, $p(i)$ is a constant. Then we have

$$p(o|i) \propto p(i|o)p(o).$$

We use $p_\phi(i|o)$ as importance weight (IW) for the gradient of individual action-value function/policy to make sure that the estimate of the expected gradients based on $p(o)$ is unbiased from that based on $p(o|i)$.

$$\mathbb{E}_{o \sim p(o|i)} \nabla_{\theta_i} J(\theta_i) \propto \mathbb{E}_{o \sim p(o)} p_\phi(i|o) \nabla_{\theta_i} J(\theta_i).$$

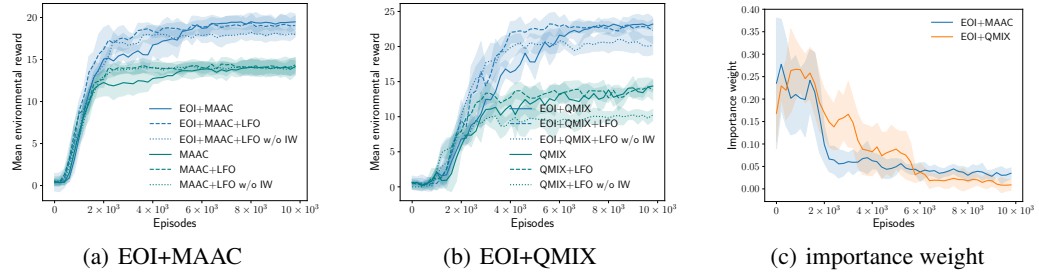

|  |  |  |
| --- | --- | --- |
| (a) EOI+MAAC | (b) EOI+QMIX | (c) importance weight |

Figure 11: Learning curves of EOI with LFO on top of MAAC (a) and QMIX (b), and the curve of importance weight during the training (c) in Pac-Men.

Figure 11(a) and 11(b) show the learning curves of EOI (with both PD and MI) with unbiased learning from others' experiences (LFO) and without importance weight respectively in Pac-Men. LFO greatly accelerates the learning of EOI since the agents have access to more experiences, and it converges to the similar performance with EOI since the gradient estimation is unbiased. Applying LFO to MAAC and QMIX also accelerates the convergence. Without important weight, LFO leads to lower mean reward, because the gradient estimate using others' experience is biased, which would harm the learning. To evaluate how much knowledge is learned from others, in Figure 11(c) we plot the curves of mean important weight on other agents' experiences $\mathbb{E}_{i,o}[\frac{1}{n-1}\sum_{j\neq i}p_\phi(i|o_j)]$, where $i$ and $j$ are the agent indexes before rearranging. At the beginning, as the agents are exploring the environment, the mean importance weight is high, which indicates that the agent learns a lot from others' experiences. In later phase, the mean importance weight becomes very low, because individuality has emerged and the experiences of agents are very different, the agent cannnot benefit from others' experiences. In the experiment, $|\omega|/|\Omega|$ is set as a constant $0.2$. However, enlightened by the trend of the mean importance weight, we could also adjust $|\omega|/|\Omega|$ according to a pre-defined schedule, *e.g.*, time-decaying or adaptive methods related to the mean importance weight.

## A.2 MATHEMATICAL FORMULATION

Let $\boldsymbol{\theta}$ and $\phi$ denote the parameters of the joint policies and the probabilistic classifier, respectively. Then, the whole learning process corresponds to the following bi-level optimization:

$$\max_{\boldsymbol{\theta}\in\Theta} \quad J(\boldsymbol{\theta}, \phi^*(\boldsymbol{\theta}))$$
$$s.t. \quad \phi^*(\boldsymbol{\theta}) = \arg\min_{\phi'\in\Phi}\mathcal{L}(\phi', \boldsymbol{\theta}),$$

where $J$ is the RL objective with intrinsic reward, $\mathcal{L}$ is the loss function of the probabilistic classifier, and $\phi$ is an implicit function of $\boldsymbol{\theta}$. Therefore, to solve this optimization, we can iteratively update $\boldsymbol{\theta}$ by

$$\frac{\mathrm{d}J(\boldsymbol{\theta}, \phi^*(\boldsymbol{\theta}))}{\mathrm{d}\boldsymbol{\theta}} = \left.\frac{\partial J(\boldsymbol{\theta}, \phi)}{\partial\boldsymbol{\theta}}\right|_{\phi=\phi^*(\boldsymbol{\theta})} + \left.\frac{\mathrm{d}\phi^*(\boldsymbol{\theta})}{\mathrm{d}\boldsymbol{\theta}}\frac{\partial J(\boldsymbol{\theta}, \phi)}{\partial\phi}\right|_{\phi=\phi^*(\boldsymbol{\theta})}$$

where

$$\frac{\mathrm{d}\phi^*(\boldsymbol{\theta})}{d\boldsymbol{\theta}} = -\left.\left(\frac{\partial^2\mathcal{L}(\phi, \boldsymbol{\theta})}{\partial\phi\partial\phi^T}\right)^{-1}\left(\frac{\partial^2\mathcal{L}(\phi, \boldsymbol{\theta})}{\partial\phi\partial\boldsymbol{\theta}^T}\right)\right|_{\phi=\phi^*(\boldsymbol{\theta})}$$

which is obtained by the implicit function theorem. In practice, the second-order term is neglected due to high computational complexity, without incurring significant performance drop, such as in meta-learning and GANs. Therefore, we can solve the bi-level optimization by the first-order approximation with iterative updates:

$$\phi_{k+1} \approx \arg\min_\phi \mathcal{L}(\phi, \mathcal{B}_k)$$
$$\boldsymbol{\theta}_{k+1} = \boldsymbol{\theta}_k + \zeta_k\nabla_{\boldsymbol{\theta}}J(\boldsymbol{\theta}, \phi_{k+1}).$$

## A.3 HYPERPARAMETERS

The hyperparameters of EOI and the baselines in each scenario are summarized in Table 1. Since QMIX and MAAC are off-policy algorithms with replay buffer, we do not need to maintain the buffer

Table 1: Hyperparameters

| Hyperparameter | Pac-man | Windy Maze | Firefighters | 8m |
|---|---|---|---|---|
| runs with different seeds | 5 | 10 | 5 | 3 |
| horizon ($T$) | 30 | 15 | 20 | 120 |
| discount ($\gamma$) | | 0.98 | | |
| batch size | | 128 | | 64 |
| replay buffer size | | $2 \times 10^4$ | | $2 \times 10^5$ |
| # MLP units | | $(128, 128)$ | | $(64, 64)$ |
| actor learning rate | | $1 \times 10^{-3}$ | | - |
| critic learning rate | | $1 \times 10^{-4}$ | | - |
| QMIX learning rate | | $1 \times 10^{-4}$ | | |
| MLP activation | | ReLU | | |
| optimizer | | Adam | | |
| $\phi$ learning rate | | $1 \times 10^{-3}$ | | |
| $\alpha$ in QMIX | | 0.05 | | |
| $\alpha$ in MAAC | | 0.2 | | |
| $\beta_1$ | | 0.04 | | 0.01 |
| $\beta_2$ | | 0.1 | | 0.01 |
| $\Delta t$ | | 4 | | |

$\mathcal{B}$ but build the training data from the replay buffer $\mathcal{D}$. For EDTI, ROMA, HC, and HC-simul, we use their default settings. In the experiments, the agents do not share the weights of neural network since parameter sharing causes similar agent behaviors. All the curves are plotted using mean and standard deviation.

## A.4 ADDITIONAL RESULTS IN PAC-MEN

In Figure 12, we compare EOI with HC and ROMA in Pac-Men. EOI converges to a higher reward than these two methods for behavior diversification, showing that EOI is more conducive to the performance improvement.

To investigate the performance of EOI in the scenarios where individuality is not necessary, we perform the experiments on an easy version of Pac-Men, where the dots are only distributed in the bottom room, illustrated in Figure 13 (*left*). To complete this task, the agents do not need to go to different rooms. As depicted in Figure 13 (*right*), EOI+QMIX converges to the same mean reward with the QMIX, which verifies that EOI does not conflict with the success of task where agent individuality is not necessary.

In Figure 14(a), we test EOI+MAAC with different $\alpha$ to investigate the effect of $\alpha$ on the emergent individuality. When $\alpha$ is too large, the agents will pay much attention to learn the individualized behaviors, which harms the optimization of the cumulative reward. However, the results of EOI+MAAC with different $\alpha$ are all higher than that of the vanilla MAAC without EOI ($\alpha = 0$), which shows that the proposed intrinsic reward could encourage individualized policies for better cooperation. In Figure 14(b), we apply EOI to PPO, where each agent is controlled by an independent PPO. EOI+PPO outperforms the version without EOI, which verifies the effectiveness of EOI on independent learning algorithms. Intuitively, maximizing the KL-divergence between agents $\sum_{j \neq i} KL(p_i(a|o)||p_j(a|o))$ for agent $i$ as a regularization could also encourages the individuality. In Figure 14(c), we compare EOI against the KL-divergence and find that KL-divergence does not outperform the vanilla MAAC,

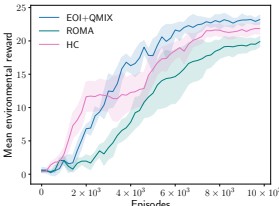

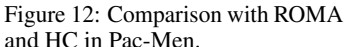

Figure 12: Comparison with ROMA and HC in Pac-Men.

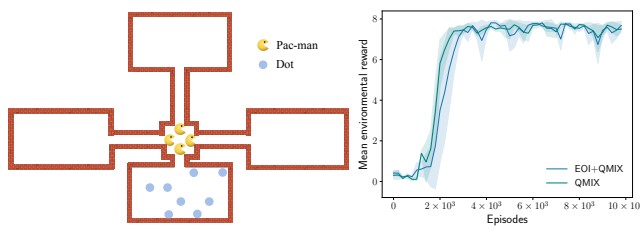

Figure 13: The easy version of Pac-Men (*left*) and learning curves (*right*).

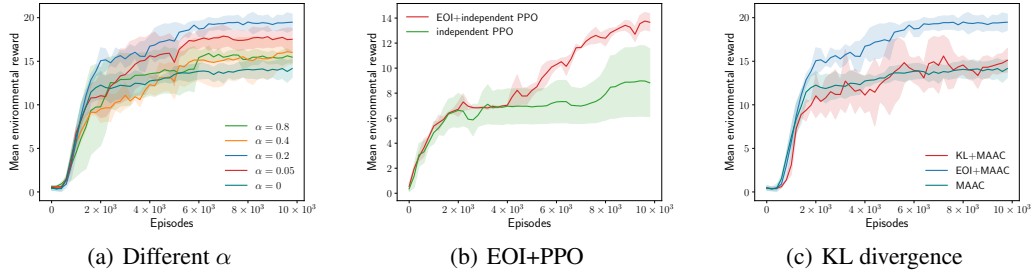

(a) Different $\alpha$        (b) EOI+PPO        (c) KL divergence

Figure 14: Learning curves of EOI+MAAC with different $\alpha$ (a), learning curves of EOI+PPO (b), and learning curves of KL divergence+MAAC (c) in Pac-Men.

and the learning of KL-divergence is not stable. We think that the KL-divergence term is hard to optimize with more than two agents.

### A.5 Similarity and Difference Between EOI and DIAYN

DIAYN is proposed to learn diverse skills with single-agent RL in the absence of any rewards. It trains the agent by maximizing the mutual information between skills ($Z$) and states ($S$), maximizing the policy entropy, and minimizing the mutual information between skills and actions ($A$) given the state. The optimization objective is

$$\begin{aligned}
&\mathrm{MI}(S; Z) + \mathcal{H}(A|S) - \mathrm{MI}(A; Z|S) \\
&= \mathcal{H}(Z) - \mathcal{H}(Z|S) + \mathcal{H}(A|S) - (\mathcal{H}(A|S) - \mathcal{H}(A|Z, S)) \\
&= \mathcal{H}(Z) - \mathcal{H}(Z|S) + \mathcal{H}(A|Z, S).
\end{aligned}$$

To maximize this objective, in practice DIAYN gives the learning agent an intrinsic reward $\log q(z|s) - \log p(z)$, where $q(z|s)$ approximates $p(z|s)$ and $p(z)$ is a prior distribution. EOI gives each agent an intrinsic reward of $p(i|o)$. Let agents correspond to skills and observations correspond to states, then EOI has a similar intrinsic reward with DIAYN. However, simply adopting the intrinsic reward in MARL, which can be seen as EOI without the regularizers, cannot provide strong intrinsic reward signals for the emergence of individuality. The two regularizers, which capture the unique characteristics of MARL, strength the reward signals and promote the emergence of individuality in MARL.

