# OpenReview forum: "The Emergence of Individuality in Multi-Agent Reinforcement Learning"
_ICLR.cc/2021/Conference — Reject_

### Official Review · AnonReviewer4 · 2020-10-17
**Method is shown to be effective in experiments, but there are concerns about positioning relative to existing work and correctness of implementation**

**Rating:** 6
**Confidence:** 5

**Review:**

This paper contributes a method based on reward-shaping to encourage the emergence of distinct agent behaviors in fully-cooperative multi-agent reinforcement learning (MARL), within the paradigm of centralized training with decentralized execution. They propose to learn a classifier that predicts agent identity given the agent's observation, and use the classifier probability as an intrinsic reward that is added to the environment reward. As such, there is a positive feedback loop whereby agents are rewarded for being distinguishable and hence generate data that is easier for the classifier to predict correctly. They provide two regularizers to help kick-start the positive feedback loop when agents' policies are similar at initialization. The paper implements the method, called EOI, on top of two base MARL algorithms. Using three illustrative environments where optimum team performance requires distinct behaviors among agents, the paper shows that 1) EOI outperforms the two base algorithms and other baselines that address exploration and diversity; 2) the two regularizers improve performance and help the intrinsic reward to serve as an inductive bias; 3) agents do perform distinct behaviors. They also show comparable performance on a benchmark StarCraft II micromanagement task.

I give a rating of 5 for the following reasons. Classification-based intrinsic reward have been used for diversification in MARL in previous work (Lee et al., 2020, Yang et al. 2020), but this paper provides a different formulation along with effective heuristics to speed up the initial training of the classifier. The experiments are clear in showing the benefit of EOI and EOI + regularizers when built on top of the two base methods.
However, there are main points of concern. I am willing to raise my score if the main points below are sufficiently addressed.

This paper incorrectly positions itself against the previous works Lee et al. (2020) and Yang et al. (2020). In section 2, the authors say "none of these studies appropriately link behavior diversification to the success of the cooperative task". However, in section 3, the authors say that "EOI directly correlates individuality with the task by intrinsic reward", making it seem that EOI differs from Lee and Yang in that regard. This is not true. The intrinsic reward in EOI is the probability of correct classification of an agent identity. The intrinsic reward in Lee and Yang are based on correct classification of a latent skill variable. Neither agent identity nor a latent skill variable are directly linked to task success. The intrinsic rewards in EOI and in these previous work encourage the emergence of distinguishable behavior; none of them have any direct link to task success. The authors either need to explain their claim in more detail, or amend the paper.

Another main concern is about the case where EOI is built on top of QMIX. The authors define $Q^p_i( o_i, Q^a_i(o_i))$, which takes in the vector of individual action-values $Q^a_i(o_i)$, and define a TD error loss with the intrinsic reward $p(i|o)$. In the first equation in Section 3.3, the TD error serves purely to _estimate_ the expected cumulative intrinsic reward under the current policy. It does not find some Q-function that _optimizes_ the expected cumulative intrinsic reward. However, the initial formulation on page 3 (where the total reward is the environment reward plus the intrinsic reward) implies that the agents should maximize the expected cumulative intrinsic reward as well. Therefore, it is not clear at all how the intrinsic reward is being maximized in the QMIX case.

Points that did not affect the decision, but where the paper can be improved:
1) All equations should be numbered for easier reference.
2) First sentence in abstract needs a language check.
3) The introduction says "...agents tend to visit their own familiar observations". It seems that this has a negative impact on the exploration by an arbitrary agent. Authors should include more discussion.
4) Section 3.3, a more succinct way is to say that assigning the intrinsic reward to the specific agent bypasses the multi-agent credit assignment problem.
5) In Section 3.3, it is not clear what the authors mean by saying that the individual value (in QMIX) has no "actual meaning". Please be more precise.
6) At the end of Section 3.3, the authors say that due to off-policy learning, $p_{\phi}(i|o)$ is recomputed for each sampled batch. Is it recomputed purely because the current classifier is presumably more accurate? Is there any potential problem with doing so, in the context of off-policy learning?
7) All plots need thicker line width, including the legend. Colors need to be more distinguishable from one another. Currently, it is hard to see which curves correspond to which algorithms.

---
Edit: score increased from 5 to 6

---

> ### Author Response · Authors · 2020-11-16
> **Responses to Review #4**
>
> Our claims about the previous methods Lee et al. (2020) and Yang et al. (2020) are inaccurate. Actually, neither agent identity nor a latent skill variable is directly linked to task success. However, the two previous methods are hierarchical structures, where the sub-skills of each agent are pre-trained independently without coordinating with the skills of other agents, and then a controller is trained for task success to coordinate the agents. EOI is instantiated on CTDE methods, which means the emergence of individuality is accompanied by jointly maximizing the centralized Q value; thus, individuality is closely linked to task success. We have removed the claim in the revision.
>
> Actually, $Q_i^p(o_i,Q_i^a(o_i))$ is the estimation of the expected cumulative intrinsic reward under the current policy. To optimize the expected cumulative intrinsic reward, we update the individual action-values $Q_i^a(o_i)$ by adding an auxiliary gradient to $\theta_i$, the second term of the second equation in Section 3.3: $- \alpha\frac{\partial Q^p_i(o_i,Q^a_i(o_i;\theta_i))}{\partial Q^a_i(o_i;\theta_i)} \frac{\partial Q^a_i(o_i;\theta_i)}{\partial \theta_i}$. It is a DDPG-like update, where $\theta_i$ is adjusted by just gradient ascent to maximize $Q_i^p$. As pointed by QMIX paper, the individual value function $Q_i^a$ in QMIX does not estimate a real expected return, so the value has no actual meaning. Therefore, adding the auxiliary gradient to $\theta_i$ does not break any constraints.
>
> EOI does not limit the exploration; instead, it could help the multi-agent exploration. Once an observation is familiar to one agent, other agents would get low intrinsic rewards at that observation, promoting other agents to explore different observations.
>
> The intrinsic reward model $ p_{\phi}(i|o) $ is updating along with the learning process; thus, the recorded intrinsic reward in the replay buffer will be outdated. There might be two experiences with the same observations and actions having two different recorded intrinsic rewards, which would harm the off-policy learning. The recomputed intrinsic reward could avoid this problem, although it might cause the moving target problem. However, the moving target is usually observed in Deep RL, and it is less severe than the outdated experience.

---

> > ### Comment · AnonReviewer4 · 2020-11-20
> > **Authors addressed concerns to some extent. I recommend further revision of the text.**
> >
> > I appreciate the authors' response regarding my two main concerns. I have raised the score. I have follow-up suggestions on improving the clarity of the paper on these two points:
> >
> > 1. Positioning with respect to the previous works (Lee et al. and Yang et al.) on discovering skills.
> > By my understanding, the main conceptual difference between EOI and previous work is that agent identity is the predicted label in EOI (and obviously an agent can only have one identity label), while the previous work discover a set of skills that can be shared among agents, so that any agent can choose any skill (and more than one skill) at different time points in an episode. I disagree with the author's characterization of the previous works that "sub-skills of each agent are pre-trained independently without coordinating with the skills of other agents". The skills are not independent, because the objective function encourages the emergence of skills that generate significantly different behavior. There is implicit coordination, because these previous methods also conduct centralized training with a single extrinsic reward.
> > 1b. In the revised paper, there is the sentence "Unlike existing work on behavior diversification, EOI directly correlates individuality with the task by intrinsic reward." This sentence is likely to cause confusion. If the authors mean that the previous work don't correlate _individuality_ with task success, then yes, of course they don't. However, the previous work do correlate their _skills_ with task success due to centralized training, and this is the same mechanism that enables EOI to correlate individuality with task success. I encourage the authors to make these subtle points clearer in their revision.
> >
> > 2. My concern is addressed by the author's explanation that equation (7) is a DDPG-style update that does in fact optimize the intrinsic reward. So this means that there is an induced continuous action space, since the individual action-value vector $Q^a_i(o_i;\theta_i)$ is used as an _action_ input. I urge the authors to include this explanation and cite DDPG in their text.

---

### Official Review · AnonReviewer1 · 2020-10-21
**Important problem, but unsure if this bias makes sense**

**Rating:** 5
**Confidence:** 4

**Review:**

This paper tackles the problem of exploration in multi-agent RL,
formulated as a Dec-POMDP. The authors propose to shape the reward
using the output of a classifier that tries to determine which agent
saw a particular observation. Moreover, the authors propose some
regularization schemes to "break ties" early on in the training. Then,
it is shown that the proposed reward shaping term can be integrated
into two popular MARL algorithms that use centralized training: MAAC
and QMIX. The integration into MAAC is relatively straightforward
because MAAC uses independent critics, whereas the integration into
QMIX is more involved due to the Q-function mixing step.

I believe this is a very important problem being tackled by the
authors. Even in single-agent RL, exploration is a major issue; this
only gets even more apparent in the multi-agent setting. I believe
that the paper was well-written, and the Introduction made it very
clear what exactly the paper's contributions were, which I greatly
appreciate.

Unfortunately, I'm not sure that I believe that the proposed bias is
truly useful. I understand the intuition: we want agents to
"specialize" their observations, such that it is easy to predict which
agent is receiving any particular observation. However, won't this be
counterproductive in most practically interesting domains? For
example, consider a team of robots working together in a factory or a
household; they will constantly be changing their environment as they
take actions toward operating the factory equipment or cleaning the
kitchen, which means the observations they receive will always be
changing. But the proposed reward shaping mechanism in this paper
would "fight" against this progress, because it would encourage the
agents to engage in trivial behavior just to be able to see the same
observations over and over. In my mind, it would be important to
consider the dynamics of how the classifier p(i | o_i) is changing
over time.

Another option could be to simply encourage the agents to learn
different policies, maybe measured via KL-divergence of P_{agent 1}(a
| s) and P_{agent 2}(a | s). I believe that this bias would be
sufficient to solve the example presented in Figure 1? Does this bias
seem reasonable, and if so, how does it compare against EOI?
Basically, more broadly, I would have liked to see experimental
comparisons that convince me that EOI is the *correct* bias to use
versus other natural biases, whereas the current experiments only seem
to compare against non-shaping methods like ROMA and HL.

Some other questions:

1. How significant is the fact that you are ignoring second-order
effects in solving the bi-level optimization in Appendix A.2? Have the
authors conducted preliminary experiments to prove that it doesn't
make much difference? I understand that this practice is standard in
methods like MAML, but it would be nice to verify that it doesn't
matter in this setting either.

2. Looking at the shaped reward computation, r + alpha * p(i | o_i),
it seems like if the classifier were naive and simply outputted a
uniform distribution, you would still be giving positive intrinsic
reward in that case. Might it make more sense instead to consider the
divergence between p(i | o_i) and the uniform distribution?

---

> ### Author Response · Authors · 2020-11-16
> **Responses to Review #1**
>
> The behaviors of trained agents are influenced by both the environmental reward and the intrinsic reward. If the agents just always engage in trivial behavior to see the same observations, the environmental reward is very low, and this behavior would be changed by optimizing the environmental reward. In the task Windy Maze, no EOI agent learns to trivially stay at the bottom of the T-shaped maze, although it is a solution to maximize the intrinsic reward.
>
> The KL-divergence of policies is hard to optimize when the agent scale is large and not compatible with deterministic algorithms, e.g., QMIX. We performed additional experiments to compare EOI against the KL-divergence based policies, which optimizes $\sum_{j\neq i}^{}\mathrm{KL}(p_i(a|o)||p_j(a|o))$ for agent $i$ as a regularization. As illustrated in Figure 14(c), KL-divergence does not outperform the vanilla MAAC, and the learning of KL-divergence is not stable. Moreover, the baselines EDTI and HC are both reward-shaping methods. The intrinsic reward in EDTI promotes the multi-agent exploration based on influence, and the intrinsic reward in HC encourages each agent to learn different skills. The experimental results show that EOI is a better bias for individuality.
>
> It is reasonable to give positive intrinsic reward when the output of the classifier is uniform. Considering a two-agent case, $p(\cdot |o_1) = [0.5,0.5]$ and $p(\cdot |o_2) = [0.4,0.6]$. For agent 1, $o_1$ is the more familiar observation, and giving a reward $p(1|o_1)=0.5$ to agent 1 at $o_1$ could encourage the agent 1 to go to $o_1$, which eventually helps the emergent individuality.
>
> The second-order term in bi-level optimization is investigated in [1]. It is shown empirically that the performance gain is marginal but the cost (training time) increases 2-3 times. Therefore, we did not consider the second-order term in the experiments.
>
> [1] Liu et al., DARTS: Differentiable Architecture Search, ICLR'19.

---

### Official Review · AnonReviewer3 · 2020-10-27
**The topic of this paper is fascinating, but, in my opinion, the authors do not really provide an insightful model about the emergence of individuality**

**Rating:** 4
**Confidence:** 4

**Review:**

The paper discusses an analysis of the emergence of individuality in a multi-agent system based on reinforcement learning. The emergence of individuality is based on intrinsic reward. The intrinsic reward is assigned to each agent by forcing them to be "different" according to a given distribution. The reviewer struggles to see this mechanism particularly insightful. This does not appear to the reviewer as emergence of individuality; his looks more like the emergence of different behavior (which can be seen as individuality, but it is more like forcing different behavior, not individual behavior, as interpreted normally when we study societies in my opinion).

Strength:

- According to the reviewer, any study that sheds a light on human phenomena using reinforcement learning is fascinating.

Weakness:

- The model used for studying the emergence of individuality is not convincing in my opinion. In fact the authors essentially impose the emergence of individuality from outside by giving a reward for being different.

- The reviewer wonders if the emergence of individuality at the end of the day might actually emerge just by considering individual learning models. I would say that this is what you observe in many multi-agent systems with individual learning models. What is the main different in this case? This is not completely clear to the reviewer.

- The choice of the mechanisms (MAAC, QMIX, etc.) is not sufficiently discussed by the authors.

- The goal of the evaluation is unclear: the goal of this evaluation is not the emergence of individuality in my opinion.

In general, I would say that the question that the authors try to address is indeed a fascinating one. However, the reviewer really struggles to understand the choice of the mechanism selected by the authors. The reference to biology are not convinced. The authors say: "Biologically, the emergence of individuality is attributed to innate characteristics and experiences. However, as in practice RL agents are mostly homogeneous, we mainly focus on enabling agents to develop individuality through interactions with the environment during policy learning". Is it not something that happens if you have individual models in any case? It seems to me that the individuality/differences are artificially imposed on the system itself.

Overall, the contribution of the paper is not significant in my opinion, since unfortunately the work does not really provide insights about emergence of individuality, which is the main stated goal of the paper.

Questions:

1. How do you map the mechanism used for the emergence of individuality to real situations? This appears in a sense as your goal, but it is difficult to see how it is possible to interpret the results of the simulation considering it as a simulation of a real-world society.

2. In the Introduction the authors say "Analogically, the emergence of individuality should also be essential for multi-agent cooperation". I would claim that this is difficult to prove by analogy as stated by the authors.

3. The goal of the evaluation is not completely clear. In fact you simulate a variety of games, etc. However, it is unclear how this  is related to the fundamental problem of the paper, which is the emergence of individuality.

4. Why do you use "positive distance"? What is the motivation of this choice?

5. The discussion about QMIX is not completely clear. Why is it necessary? How do you link this with the goal of the paper? Is there any clear mapping with real-world situations?

---

> ### Author Response · Authors · 2020-11-16
> **Responses to Review #3**
>
> For individual learning models, as explained in the first paragraph of Section 3.1, agents may easily converge to local optimum in the scenario where the division of labor is desired, and thus they learn similar behaviors (e.g., both agents collect item b in Figure 1). Moreover, The difference between individual learning models and EOI can also be verified by our additionally performed experiments. As illustrated in Figure 14(b), EOI+PPO largely outperforms PPO (individual learning models). This shows that different behaviors do not emerge naturally in individual learning models.
>
> EOI could be built on any multi-agent cooperation methods. MAAC and QMIX are respectively typical policy gradient method and value decomposition method, so we select them as backbone algorithms.
>
> The goal of the emergence of individuality is to get higher environmental rewards, which means better cooperation. In the experimental scenarios, the policies with individuality could achieve higher rewards. Therefore, the sum of environmental rewards is the most important metric of individuality.
>
> Some sentences, e.g., "Biologically, the emergence of individuality is attributed...." are stated to conceptually motivate this work and help readers to easily understand the intuition behind this work.
>
> Positive distance is adopted from the triplet loss, which is proposed to learn identifiable embeddings. The positive distance minimizes the intra-distance between the observations with the same “identity”, encouraging the observations of one agent easily identified from that of other agents.

---

### Official Review · AnonReviewer2 · 2020-10-30
**Solve Dec-POMDP problem based on MAAC and QMIX. Interesting and timely work~**

**Rating:** 6
**Confidence:** 5

**Review:**

This article solves the Dec-POMDP problem, based on MAAC and QMIX algorithms. When the agents are homogeneous and optimized only through team rewards, it is easy to learn similar policies for each agent. This makes the multi-agent algorithm finally converge to a locally optimal joint policy. If each agent can complete different parts of the overall goal, the joint policy that converges to is obviously better. Many current works model the above problems as task assignment or role assignment problems. However, the agents with different roles in these works are basically rule-based, and the tasks are all manually defined. There are also some works that unsupervisedly generate different strategies and roles by introducing diversity constraints, but the generation process has nothing t o do with the task. In order to solve the shortcomings of the above methods, this paper proposes a MARL method based on reward shaping to encourage the division of labor between agents, and at the same time introduces two regularize terms to solve the problem of too similar agent policies at the initial training stage. At the same time, reward shaping and reinforcement learning are optimized simultaneously, forming a bi-level optimization problem. The paper also designed three tasks that emphasize division of labor to verify the effectiveness of the algorithm.
1. Neither the MAAC nor the QMIX algorithm on which the paper is based has good scalability. Although the independent learning algorithm is simple, it can achieve better performance on many tasks and has good scalability. This paper should additionally use independent learning algorithms as baselines, and apply the intrinsic rewards proposed in this paper to independent learning algorithms.
2. The three tasks used to verify the algorithm in this paper are all specially designed, with a strong emphasis on division of labor. I think the paper should additionally explain the limitations of the algorithm, such as which scenarios will be more effective and which scenarios will limit the learning ability of the agent.
3. The optimization process of bi-level problems is very unstable. The algorithm proposed in this paper contains many hyperparameters, and the sensitivity of the algorithm to hyperparameters should be shown in the experimental part.

---

> ### Author Response · Authors · 2020-11-16
> **Responses to Review #2**
>
> We agree that the independent learning algorithms, e.g., independent PPO or DQN, could be applied to large-scale multi-agent environments. However, in our tasks, the agents could only get a limited local observation, and the estimation of the global state is strongly related to the joint action, so it is hard to train the independent agents. Nevertheless, we performed additional experiments on applying EOI to independent PPO. As illustrated in Figure 14(b) in the revision, EOI could also promote the performance on independent learning algorithms.
>
> EOI could help the tasks where the division of labor is strongly required, and the agents would trap into the local optimum of similar policies. For the scenarios where the division of labor is not necessary, as shown in Figure 13, EOI does not conflict with the success of the task. If the optimal solution is to learn similar policies, e.g., the agents should go to the same target, EOI should not be considered, and parameter sharing could work well.
>
> The most important or sensitive hyperparameter in EOI is $\alpha$ in $r+\alpha p\left(i \mid o_{i}\right)$. As suggested, we tested different $\alpha$ in Figure 14(a) in the revision. When $\alpha$ is too large, the agents will pay much attention to learn the individualized behaviors, which harms the optimization objective. However, the results of EOI+MAAC with different $\alpha$ are all better than that of the vanilla MAAC (without EOI, $\alpha = 0$), which shows that the proposed intrinsic reward could encourage individualized policies for better cooperation.

---

### Author Response · Authors · 2020-11-16
**We have posted the responses to each reviewer and will continuously address any further comments.**

We hope the responses could address the comments of all the reviewers. Besides, we have performed additional experiments as suggested by some reviewers, and the results are included in the *revision*.

We look forward to fully discussing the comments and concerns of the reviewers. Hope they could fully understand the merits of the paper.

---

### Decision · Program_Chairs · 2021-01-07
**Final Decision**

**Decision:**

Reject

**Comment:**

This paper introduces a method to increase diversity/individuality of agents in a MARL setup, based on intrinsic rewards coming from a classifier over behaviours.

Reviewers tend to agree that this is an important/interesting problem, which is related to exploration, a central problem in reinforcement learning. Several reviewers point out that the paper is well written. I appreciate that the authors have been responsive to reviews and have answered and/or addressed several points of concern of the reviewers. The proposed method performs well on the experiments carried out.

Reviews still point out several things that could be improved. The experiments mostly report reward curved, and only few results are actually clearly pointing out the individuality between agents. The fact that this method outperforms the baselines is good, but does not prove individuality and may simply be due to the authors spending more time on the tasks, or other undiscovered phenomenon.
A reviewer is concerned that this extra reward could encourage trivial behaviours, and it seems clear that it will if the relative weight of the intrinsic reward is too high. This should be discussed more.
Finally, a reviewer points out that classifier-based intrinsic reward for diversity already exists in published works and that this paper is incremental work.

The average score for this paper is very close to the acceptance threshold, but based on the reviews I recommend to reject this paper for ICLR 2021. I am confident that when the authors address further the reviewers concerns and improve the experimental results, this paper will be published in a future venue.